# OpenReview forum: "The Best of Both Worlds: Bridging Quality and Diversity in Data Selection with Bipartite Graph"
_ICLR.cc/2025/Conference — Submitted to ICLR 2025_

### Official Review · Reviewer_scTA · 2024-10-18

**Soundness:** 3
**Presentation:** 3
**Contribution:** 2
**Rating:** 6
**Confidence:** 4

**Summary:**

This paper introduces a method to filter data for supervised fine-tuning of LLMs.  Given a budget of a certain number of instruction+response pairs on which to fine-tune, this paper proposes that we should prioritize selecting both high-quality instructions (assessed with some kind of quality metric) and diverse instructions (by selecting new unique n-grams).  Prior work has focused on either quality or diversity in isolation.  The approach is implemented using a data structure called GraphFilter, which links n-grams to the sentences that they occur in.  The algorithm proceeds in a greedy manner, selecting instructions that have the highest product of quality times diversity, and then removing all selected n-grams from future assessments of diversity (meaning instructions with those n-grams will now be weighted lower in the next iteration).  The proposed method is shown to improve over various baselines, and ablations show that using both the quality score and the diversity score together are better than using either in isolation.  Another interesting experiment shows that the value of quality is highest with smaller budgets, whereas with larger budgets, the importance of diversity grows.

**Strengths:**

For the most part, the paper is clearly-written and well organized.  It’s relatively easy to understand what has been done, and, many of the things that were done are well-motivated and sound.  Figure 2 and Algorithm 1 add to the clarity.  The ablations are presented well in the Tables.  The main claim: that we need quality and diversity together to select a good dataset for supervised fine-tuning (SFT), is well-supported by the experiments.  It’s a fairly reasonable idea, and seems potentially beneficial to others doing supervised fine-tuning.

**Weaknesses:**

The overall motivation for a SFT data budget is not perfectly motivated.  What controls the size of this budget?  Do we see diminishing/negligible returns as we FT over more and more data?  Could downstream accuracy actually suffer as we move to lower-quality data?  Or are we just limited by our own compute budgets for doing the supervised fine tuning?  It would have been useful to motivate this further in the paper, as the exact motivation can affect the reviewer evaluation of what experiments should have been run, etc.  Right now, it’s not clear whether filtering data generated by an LLM for SFT is as important as filtering data scraped from the web for pretraining.

I am confused about the use of TFIDF in this context.  Normally, TF is the term frequency *within a document*, and IDF is the inverse of the frequency of the term *across documents*.  It’s a good metric for finding keywords in a specific document because “important” terms are frequent in the given document, but less frequent across documents.  But here, TF seems to mean the *global* term frequency, while IDF represents the inverse frequency *across sentences*.  I struggled to think of examples where this is useful.  For many terms, I would expect them to occur at most once in each sentence, so it seems to me that TF and DF would be very similar for most terms, and thus the overall “diversity score” scales with x log(1/x), where x is the frequency of the term.  I do not expect this score to encourage diversity; indeed, I think diversity arises from removing the n-grams from consideration that have already been used in instructions – so on subsequent iterations, sentences containing previously-unseen n-grams will get higher scores.  It is not clear – and ablations do not test – if using what-is-called TFIDF provides a benefit over, say, counting each N-gram based purely on its frequency, or even giving each N-gram a score of 1.  Since both using “TFIDF”, and calling it that, seem incorrect to me, and this score is more complicated than other approaches, it’s hard for me to recommend using this algorithm to my colleagues working on supervised fine-tuning.  It’s hard to picture the paper being published with this issue.

I didn’t find Figure 1 that useful; it would be better if it illustrated how the algorithm works in a bit more detail.

Some of the claims were misleading or at least oversold.  For example, GraphFilter “captures the complex relationships between sentence and n-grams.”  From my perspective, it captures a single relationship, whether the sentence CONTAINS the n-gram… and that’s a fairly straightforward relationship.  Also, GraphFilter is “up to 61x faster than these baselines without requiring GPUs.”  But isn’t it just 61x faster than a SINGLE baseline (AlpaGasus)?  If we’re cherry-picking single baselines, how much faster is it than Random?  And isn’t it only 61x faster when not using the priority function for re-ranking?  And yet I don’t see where we report the accuracy when not using the priority function.  It’s not valid to claim it outperforms the baseline accuracy by x% and is 61x faster, when it seems these things are not JOINTLY true.  It would be like having an untested mode of GraphFilter that picks the shortest instructions and using this to justify a claim that GraphFilter is 500x faster than the baselines.  Besides, I don’t really understand what it means not to use ϕ(u), because as far as the paper says, ϕ(u) is how GraphFilter ranks sentences (Equation (4)).  Finally, the intro claims GraphFilter outperforms the baselines by “up to +2.37 for GEMMA-2-2B, +3.02 for MISTRAL-7B-V0.3, and +3.38 for LLAMA-3-8B” – but this is only over LONGEST, and that’s not even as good as RANDOM.  I am not sure the paper *intentionally* tried to mislead the reader, but after reading the introduction, the reader was *expecting* GraphFilter to beat a COMPETITIVE baseline by being BOTH 2-3% better AND 61x faster.

In terms of other ablations, since we see that unigrams to trigrams are important, it leads us to wonder, what about four-grams?  And five-grams?  Where does it level off?

I am also not sure how sound it is to report results applying GraphFilter to instructions only (and not responses).  Perhaps it would have been more fair to test each baseline being applied to either instructions only, responses only, or both, and then to report the best result across all three (assuming that was how the results were compiled for GraphFilter).

Small points (not affecting the scoring/evaluation of the paper):
- While Section 5.4 is great, you should remind everyone there are only 300K sentences in total so things will approach random (Δ_all will got to zero) as we increase the budget.
- Without using TFIDF nor SuperFilter, is it random?  So why is the result 45.61 in Table 4 and yet random gets 45.51 in the main Table 1?  Is that a typo?  Did you re-run it again?  Is it not random?

**Questions:**

Is there much variance in these results?  What about re-running multiple times in order to get error bars?

Why TFIDF and not just TF (as noted above)?  Or just giving every n-gram a score of 1.

Why Quality x Diversity?  What about Quality + Diversity?  Quality^2 * Diversity?  Quality * Log(Diversity)?  Quality^(10/log(N)) * Diversity, where N is the size of the subset.  You know what I mean?  There is some assumption underlying Quality x Diversity, and since it’s the central topic of the paper, it would be good to spell it out and test it.

---

> ### Author Response · Authors · 2024-11-19
> **Thank You for Your Review + Response (Part 1/2)**
>
> We sincerely thank you for your thorough and insightful review of our paper. Your feedback has been invaluable in highlighting areas where we can improve our work. We have carefully considered your comments and have addressed each point below.
>
> **1. Motivation**
>
> Thank you for highlighting the need for clearer motivation regarding the SFT data selection. We believe our research question is well motivated, as pointed out by Reviewer PWPP and Reviewer MtTD. Firstly, recent works have demonstrated that selecting a subset of high quality data can effectively improve the model performance, allowing for better performance compared to those models fine-tuned on the entire dataset [1,2,3]. Excessive data, especially if it's of lower quality or redundant, can slow down training and may even degrade model accuracy due to noise. Furthermore, recent LLMs are typically fine-tuned on tens of millions of SFT examples [4,5], making the SFT data selection crucial. Moreover, setting the budget of data selection is a well estabilied setup fromo prior works [2,3] and we adopt this setup in our study. We will revise the motivation section to better articulate the importance of optimizing the SFT data budget.
>
> **2. Use and Terminology of TF-IDF**
>
> We apologize for the confusion regarding our use of TF-IDF. **After carefully checking our manuscript, we find there is a typo leading to this significant confusion.** At line 270, we mentioned "Term Frequency" is the number of times n-gram v appears in the corpus". This is a typo. **In our implementation, Term Frequency is the number of times n-gram v appears in the SENTENCE**. Furthermore, we present an ablation study on the effectiveness of TF-IDF in Table 4. The variation of "SuperFilter + X" indicates disabling TF-IDF and giving each n-gram a score of 1. Our results demonstrate that incorporating TF-IDF (SuperFilter + TF-IDF) as the diversity metric can effectively improve the model performance, compared to the model giving each n-gram a score of 1 (SuperFilter + X). We will fix this error in our future revision.
>
> **3. Clarity of Figure 1**
>
> Thank you for this suggestion. We agree that Figure 1 could be more informative. We will update Figure 1 to provide a step-by-step illustration of how GraphFilter operates.
>
> **4. Clarification of Claims and Comparisons**
>
> We apologize for any confusion caused by our claims. We understand that presenting these results together without clear context can be misleading. We will revise the introduction and results sections to present speed and performance improvements separately and more transparently in our future revision.
> each
>
> **5. Inclusion of Higher-order N-grams**
>
> This is an excellent point. Exploring higher-order n-grams could provide insights into the trade-offs between computation and diversity gains. As shown in the following table, the number of n-grams increases significantly as the order of n grows, leading to substantial increase in the size of the bipartite graph. Given our current computational infrastructure, we are not able to explore the scaling effect of the order of n-grams. We will try our best to address this question in our future revision.
>
> | n | # of unique n-grams |
> |---|---------------------|
> | 1 | 0.1M                |
> | 2 | 0.8M                |
> | 3 | 1.6M                |
> | 4 | 2.2M                |
> | 5 | 2.6M                |

---

> > ### Author Response · Authors · 2024-11-19
> > **Thank You for Your Review + Response (Part 2/2)**
> >
> > **6. Filtering based on Instructions and Responses**
> >
> > Your suggestion to evaluate SOTA methods when applied solely to instructions is valuable. We believe we conduct fair comparisons in our study. **Each approach has its own optimal application**, and we adopted our baselines as suggested by prior works. We believe these approaches have been optimized by their authors. To further confirm our claim, we apply several recent SOTA methods to the instructions only and present the results using Llama-3-8B as the model backbone in the following table. We observe that the responses are highly important in determining the data quality, given the observed performance drops. It is important to note that GraphFilter also considers the quality of responses during the data selection process, because SuperFilter measures the data quality based on both the instructions and responses, as shown in Equation 2.
> > |                               | $\mu_{BENCH}$ | $\mu_{LLM}$ | $\mu_{ALL}$ |
> > |-------------------------------|---------------|-------------|-------------|
> > | Random                        | 47.75         | 41.04       | 45.51       |
> > | AlpaGasus                     |               |             |             |
> > | with instruction only         | 48.16         | 41.40       | 45.88       |
> > | with instructions & responses | 48.96         | 41.90       | 46.60       |
> > | DEITA                         |               |             |             |
> > | with instruction only         | 48.45         | 41.66       | 46.22       |
> > | with instruction & responses  | 48.78         | 41.70       | 46.42       |
> > | GraphFilter (Ours)                   | 50.55         | 42.79       | 47.97       |
> >
> > **7. Section 5.4 and Dataset Size**
> >
> > Thank you for bringing this up. We will include a note reminding readers of the dataset size and its implications on the results as the budget increases.
> >
> > **8. Results in Table 4**
> >
> > When not using TFIDF nor SuperFilter (X + X), our method is NOT reduced to random.  When not using SuperFilter nor TF-IDF (X + X), both sentence nodes and n-grams nodes are given a score of 1, as illustrated in Figure 2. In such a case, our GraphFilter selects the subset with the maximal lexical diversity. As shown in Table 4, we only observe marginal impormwhen we only consider the diversity in data selection,
> >
> > **9. Variance and Error Bars**
> >
> > Thank you for this suggestion. Given our limited computational resources, we follow the setup from previous works [2,3] and did not re-run the experiments for multiple times. We use the identical settings, such random seed and hyper-parameters, for all the approaches in our study to maximize the reproducibility and reliability of our study. We will include these additional experiments in our future revisions.
> >
> > **10. Choice of TF-IDF over TF or Uniform Scoring**
> >
> > Our intention in using TF-IDF was to balance the frequency and informativeness of n-grams. TF alone might overemphasize common n-grams, while uniform scoring treats all n-grams equally, potentially missing out on informative rare n-grams.
> >
> > **11. Formulation of the Priority Function**
> >
> > We appreciate this insightful question. We follow DEITA score [2] when defining the priority function. The multiplicative form was chosen to ensure that both quality and diversity significantly influence the priority score, with a low score in either factor reducing the overall priority. There are countless possible alternative formulations for the priority function and it is impossible to exhaustively investigate all of them. Currently, we do not have sufficient compute for investigating alternative formulations (e.g., additive, exponential) and will leave these works to the future research.
> >
> > **References:**
> >
> > [1] Zhou, Chunting, et al. "LIMA: less is more for alignment." Proceedings of the 37th International Conference on Neural Information Processing Systems. 2023.
> > [2] Liu, Wei, et al. "What Makes Good Data for Alignment? A Comprehensive Study of Automatic Data Selection in Instruction Tuning." The Twelfth International Conference on Learning Representations.
> > [3] Chen, Lichang, et al. "AlpaGasus: Training a Better Alpaca with Fewer Data." The Twelfth International Conference on Learning Representations.
> > [4] Dubey, Abhimanyu, et al. "The llama 3 herd of models." arXiv preprint arXiv:2407.21783 (2024).
> > [5] Team, Gemma, et al. "Gemma 2: Improving open language models at a practical size." arXiv preprint arXiv:2408.00118 (2024).

---

> > > ### Comment · Reviewer_scTA · 2024-11-19
> > >
> > > I thank the authors for their detailed reply.  This does address some of my main concerns.  I will adjust my score accordingly.

---

### Official Review · Reviewer_MtTD · 2024-11-03

**Soundness:** 3
**Presentation:** 3
**Contribution:** 3
**Rating:** 6
**Confidence:** 5

**Summary:**

The paper presents GRAPHFILTER, a novel method for selecting high-quality and diverse subsets of data for supervised fine-tuning (SFT) of large language models (LLMs). The method represents the dataset as a bipartite graph, linking sentences to n-grams, allowing for an efficient selection process that enhances both quality and diversity. Key contributions include:

* Bipartite Graph Model: GRAPHFILTER models sentences and n-grams as nodes in a bipartite graph, capturing n-gram diversity and allowing for efficient sentence selection.
* Priority Function for Quality and Diversity: A priority function that combines quality (measured by SUPERFILTER) and diversity (via TF-IDF for n-grams) guides the sentence selection process, ensuring a balanced and high-value subset.
* Extensive Benchmarking: Experiments on three model backbones and six benchmarks show that GRAPHFILTER outperforms nine other data selection methods, achieving better model performance and computational efficiency.

The paper demonstrates the effectiveness of GRAPHFILTER, providing insights into the selected subsets' characteristics and highlighting the importance of instruction diversity for model training

**Strengths:**

Originality

The originality of GRAPHFILTER lies in its approach to data selection for supervised fine-tuning (SFT) of large language models (LLMs). Unlike prior methods that treat quality and diversity as separate concerns, GRAPHFILTER integrates both dimensions by representing the dataset as a bipartite graph connecting sentences and n-grams. This graph structure is a novel way to capture linguistic diversity at the n-gram level while the priority function effectively balances quality and diversity within the selection process. Using a bipartite graph to represent sentence-n-gram relationships is a creative approach, setting the paper apart from conventional selection techniques that use heuristics or embeddings alone. Additionally, including a priority function that combines quality metrics like SUPERFILTER with TF-IDF for diversity provides a unique, data-driven selection mechanism that ensures relevance and informativeness of the chosen data.

Quality

The quality of the research is high, demonstrated by a thorough methodology and comprehensive experimentation. The bipartite graph model and priority function are clearly defined and logically justified, and the selection process is detailed in both algorithmic form and example illustrations. The extensive experiments cover three different model architectures and six popular benchmarks, ensuring a broad and robust evaluation. Including nine baseline approaches provides a clear and rigorous comparison highlighting GRAPHFILTER’s advantages across multiple configurations. The authors further validate the effectiveness of their design choices through ablation studies, examining how different n-gram combinations and priority functions impact performance, which adds to the paper’s methodological robustness.

Clarity

The paper is well-organized and communicates complex ideas effectively. Each approach component—dataset representation, quality and diversity metrics, and the iterative selection process—is clearly explained with diagrams and examples illustrating the graph model and selection procedure. Definitions for quality and diversity metrics and the rationale for combining them are presented straightforwardly. Using an algorithmic breakdown and a visual example to show each iteration of GRAPHFILTER enhances clarity. The experimental results are also well-presented, with tables summarizing key findings and ablation studies that provide deeper insights into specific aspects of the approach.

Significance

GRAPHFILTER is significant because it offers a scalable, efficient solution to the data selection problem in LLM fine-tuning, a critical need given the vast datasets and resources involved in training. By balancing quality and diversity, GRAPHFILTER creates more representative and effective subsets, which can improve model generalization without requiring extensive computational resources. This can potentially enhance SFT for LLMs in domains where data quality and diversity are essential, such as multilingual NLP and domain-specific applications. Moreover, the method’s efficiency is demonstrated by its ability to run on CPUs while achieving faster runtimes than GPU-dependent baselines, making it accessible for more resource-constrained environments. The open-sourcing of GRAPHFILTER and its accompanying data selection scripts provides a valuable tool for future research in data-efficient model training.

**Weaknesses:**

1. Limited Adaptability Beyond Current Quality Metrics
Weakness: The method relies heavily on the SUPERFILTER metric for quality assessment, which may limit its generalizability, particularly in domains or languages not well-represented by this metric. This reliance could be restrictive, especially when different quality indicators are needed based on the type of LLM or domain being fine-tuned.

Recommendation: To increase adaptability, the authors could explore alternative or domain-specific quality metrics that could be seamlessly integrated into the priority function. An experiment comparing SUPERFILTER with other quality metrics (e.g., perplexity or reward-based metrics tailored to specific tasks) would provide insights into GRAPHFILTER's flexibility. Alternatively, a modular design allowing users to plug in their quality metrics could make the method more versatile and applicable across diverse datasets.

2. Lack of Comprehensive Error Analysis in Data Selection
Weakness: The paper does not provide an in-depth error analysis to examine cases where GRAPHFILTER might underperform or select less useful data, especially when quality and diversity conflict. Such analysis would be valuable for understanding the method’s limitations and refining the priority function.

Recommendation: A targeted error analysis could categorize cases where selected data may not align well with model performance. For example, the authors could examine if lower-quality but diverse data instances significantly impact model outputs or if certain n-grams introduce noise. This analysis could also help fine-tune the priority function to better manage trade-offs between quality and diversity, potentially through a weighted approach that adjusts depending on the dataset characteristics.

3. Evaluation Limited to Standard Benchmarks and Missing Real-World Scenarios
Weakness: The evaluation is limited to standard benchmarks, which, while effective for baseline comparisons, may not fully capture the method’s utility in diverse or more practical real-world scenarios. Standardized datasets often lack the complexities of domain-specific data, such as legal, biomedical, or social media text, where quality and diversity requirements vary.

Recommendation: Extending the evaluation to real-world or specialized datasets would enhance the evidence for GRAPHFILTER’s broad applicability. A domain-specific test (e.g., using a biomedical or multilingual dataset) would provide insights into how GRAPHFILTER handles more nuanced data selection needs. Additionally, incorporating a diverse data type, such as conversational or low-resource languages, could demonstrate the method’s robustness across data complexities.

4. Minimal Analysis of Computational Efficiency Gains
Weakness: While the paper notes GRAPHFILTER’s computational efficiency, it lacks a detailed analysis of where and how these gains are achieved, particularly regarding graph updates and priority re-ranking. This could limit readers’ understanding of the method’s scalability and how well it would perform on substantially larger datasets.

Recommendation: Breaking down computational costs (e.g., time spent on graph updates vs. priority re-ranking) would clarify the efficiency gains and allow for targeted optimizations. Additionally, experimenting with larger dataset subsets or more frequent updates could provide a clearer picture of how GRAPHFILTER scales with data size. If feasible, comparing runtime and memory usage across different datasets would also offer valuable insights for practical applications.

5. Overemphasis on N-gram Diversity Without Fine-Grained Control
Weakness: The method’s focus on n-gram diversity may be limited if particular n-grams or patterns are less relevant to specific tasks. In some cases, emphasizing phrase-level or domain-specific terms could be more beneficial than broad n-gram diversity, but the current setup lacks control over such granular adjustments.

Recommendation: Incorporating a mechanism to adjust the weight or importance of specific n-gram levels or types would add flexibility. For instance, allowing users to prioritize specific n-gram frequencies or filtering by linguistic features such as named entities or domain-specific terms could enhance GRAPHFILTER’s relevance for task-specific fine-tuning. This could be tested with a small-scale experiment where the diversity weight is adjusted based on target domain characteristics, such as including only highly informative bigrams in biomedical data.

**Questions:**

N/A

---

> ### Author Response · Authors · 2024-11-19
> **Thank You for Your Review + Response**
>
> We sincerely thank the reviewer for their thorough and insightful review of our paper. Below, we address the concerns and suggestions you have raised.
>
> **1. Limited Adaptability Beyond Current Quality Metrics**
>
> Our method primarily utilizes SuperFilter, which is based on perplexity, as a proxy for quality due to its effectiveness and computational efficiency. **We would like to point out that our method is compatible with any data quality metrics, and users can choose the quality metric based on their own needs.** Our method, with SuperFilter as the quality metric, achieves the best performance in our preliminary study. We demonstrate that perplexity can be used as the quality metric in Table 4. To further support our claim, we conduct additional experiments using ArmoRM and AlpaGasus as the quality metrics and Llama-3-8B as the model backbone. As shown in the following table, we observe that GraphFilter with SuperFilter achieves the best performance, and the variations of GraphFilter with Perplexity, ArmoRM, and AlpaGasus effectively outperform their quality-based counterparts.
>
> |                  | $\mu_{BENCH}$ | $\mu_{LLM}$ | $\mu_{ALL}$ |
> |------------------|---------------|-------------|-------------|
> | Random           | 47.75         | 41.04       | 45.51       |
> | Perplexity       | 48.27         | 40.28       | 45.61       |
> | ArmoRM           | 48.21         | 42.66       | 46.36       |
> | AlpaGasus        | 48.96         | 41.90       | 46.60       |
> | GraphFilter (Ours)      |               |             |             |
> | with SuperFilter | 50.55         | 42.79       | 47.97       |
> | with Perplexity  | 49.21         | 40.85       | 46.43       |
> | with ArmoRM      | 48.82         | 42.21       | 46.77       |
> | with AlpaGasus   | 49.09         | 42.22       | 46.91       |
>
> **2. Lack of Comprehensive Error Analysis in Data Selection**
>
> In Section 5.2, we conducted in-depth analysis of the selected subsets given by various data selection methods. As shown in Figure 3a, GraphFilter achieves the highest lexical diversity with regard to MTLD, and ranks second in data quality as evaluated by Skywork-Reward-Llama-3.1-8B. As visualized in Figure 3b and 3c, GraphFilter also maximizes dataset diversity. Referring to the results in Table 1, GraphFilter achieves the best balance of quality and diversity.
>
> **3. Evaluation Limited to Standard Benchmarks and Missing Real-World Scenarios**
>
> **We strongly disagree with this point. In this work, we conduct comprehensive evaluations on both standardized benchmarks and LLM-as-a-Judge benchmarks.** Those LLM-as-a-Judge benchmarks, including AlpacaEval-2.0 and MT-bench, effectively reflects the real-world scenarios and are highly correlated to human judgements.
>
> **4. Minimal Analysis of Computational Efficiency Gains**
>
> We appreciate the need for a thorough analysis of computational efficiency. To address this, we will provide a detailed breakdown of the runtime components, including time spent on graph construction, updates, and priority re-ranking in our future revision.
>
> **5. Overemphasis on N-gram Diversity Without Fine-Grained Control**
>
> We agree that providing fine-grained control over diversity aspects can enhance the applicability of GraphFilter to specific tasks. Our design aims to maximize diversity within a limited budget, but we recognize that this could impact performance on specialized tasks. Although task-specific fine-tuning is beyond the scope of our study, we believe this potential issue can be easily addressed. In our GraphFilter, we consider both quality and diversity during the data selection process. To achieve task-specific data selection, we propose two possible solutions: Firstly, we can utilize the quality metric computed by the task-specific quality estimation model for task-specific re-ranking. Secondly, we introduce a whitelist of task-specific words (e.g., a list of biomedical terminologies), where n-grams containing these words are allowed to be visited multiple times. We will include a discussion in the paper acknowledging this limitation and propose exploring strategies that allow for multiple visits to critical n-grams in future work.

---

### Official Review · Reviewer_PWPP · 2024-11-08

**Soundness:** 3
**Presentation:** 4
**Contribution:** 2
**Rating:** 6
**Confidence:** 3

**Summary:**

The paper proposes a simple yet effective method named 'GraphFilter' to efficiently select SFT data with possible CPU-only computation. GraphFilter represents a dataset as a bipartite graph connecting sentences to their n-grams, allowing it to balance the quality and diversity sentences that maximise n-gram diversity.

The paper also designs a priority function to combine quality and diversity in a multiplicative manner. The method iteratively selects high-priority sentences, updates the graph by removing covered n-grams, and recalculates priorities to adapt to the evolving dataset structure.

Experiments across three model backbones and six benchmarks demonstrate that the proposed method outperforms nine baselines in both model performance and computational efficiency. Further analysis validates the method's effectiveness and highlights the importance of balancing instruction diversity with data quality for effective model fine-tuning.

**Strengths:**

* **Significance**: The paper targets an important problem in LLM SFT, which is how to select the training data.

* **Originality**: The paper proposes an efficient and effective method to select high quality and diversity data by designing a priority score and a bipartite graph representation for the dataset.

* **Clarity**: The paper adopts various datasets and baselines. The effectiveness of the proposed method is supported by comprehensive experiments and sound ablation study.

* **Quality**: The paper is well-written. The presentation is clear and the graphs and tables are well-organised.

**Weaknesses:**

* The Related Work for data selection could be enriched. The paper may explain more the limitations of the current methods and why it is important to address these limitations. In addition, methods like S2L [1] and DiverseEvol [2], although focusing on SFT a specific task, should also be added and discussed. Otherwise, the claim for solving SFT data selection problem should be narrowed down.


* The paper divides the existing methods into two categories:  quality-focused and diversity-focused. However, they always consider other different criteria or have different explanation of quality, for example, verbosity, complexity, and helpfulness to a specific task [3,4]. The paper should also discuss why the perplexity-based score is enough to capture the quality of the data.


* In the bipartite graph, the n-gram nodes are only visited once, and the edges and nodes are removed after being visited. It indeed accelerates the data selection process and enables the method to visit as many n-grams as possible given limited budget. However, the data needed to learn a certain knowledge or gain a certain capability may not be the same. The concern is that the model may not learn well on specific tasks, as supported that in the main results the method proposed does not perform the best in all tasks.


* In Section 5.3, the paper performs a comprehensive ablation study on applying GRAPHFILTER to instructions or responses. It is notable that SOTA methods like DEITA and ARMORM are applied both on instructions and responses, while the performance for the proposed method fails to outperform SOTA methods when applied to both instructions and responses.


References:

[1] SMALLTOLARGE (S2L)- Scalable Data Selection for Fine-tuning Large Language Models by Summarizing Training Trajectories of Small Models, Yu et al., CoRR 2024.

[2] Self-Evolved Diverse Data Sampling for Efficient Instruction Tuning, Wu et al., CoRR 2023.

[3] What makes good data for alignment? A comprehensive study of automatic data selection in instruction tuning. Liu et al., ICLR 2024.

[4] Interpretable preferences via multi-objective reward modeling and mixture-of-experts. Wang et al., CoRR 2024.

**Questions:**

* Related Work like DEITA considered three dimensions: complexity, quality and diversity. DEITA tries to maintain complexity and diversity while selecting high quality data first. It would be interesting to compare the in Section 5.2 to see the improvement of GraphFilter from DEITA’s data selection.


* It is interesting to see how the performance will change if the n-gram nodes can be visited multiple times, since different task or knowledge may need different amount of training data.


* It would be interesting if the paper can try to apply SOTA methods only on instructions and do some evaluation.

---

> ### Author Response · Authors · 2024-11-19
> **Thank You for Your Review + Response (Part 1/2)**
>
> We sincerely thank the reviewer for their thoughtful and constructive feedback on our manuscript. We are pleased that you found our work significant, original, and well-presented. Your insights are invaluable, and we are committed to addressing your concerns to improve our paper.
>
> **1. Enriching the Related Work Section**
>
> We appreciate your suggestion to enhance our discussion of data selection methods in the Related Work section. We agree that incorporating a more comprehensive overview, including methods like S2L and DiverseEvol, will strengthen our paper. We will revise the manuscript to include these references and elaborate on how they relate to and differ from our approach.
>
> **2. Expanding on Quality Criteria Beyond Perplexity**
>
> You raise an important point regarding the multidimensional nature of data quality, including factors like verbosity, complexity, and task-specific helpfulness. We acknowledge that it may not capture all aspects of quality. Our method primarily utilizes perplexity as a proxy for quality due to its effectiveness and computational efficiency. **We would like to point out that our method is compatible with any data quality metrics, and users can choose the quality metric based on their own needs.** Our method, with SuperFilter as the quality metric, achieves the best performance in our preliminary study. We demonstrate that perplexity can be used as the quality metric in Table 4. To further support our claim, we conduct additional experiments using ArmoRM and AlpaGasus as the quality metrics and Llama-3-8B as the model backbone. As shown in the following table, we observe that GraphFilter with SuperFilter achieves the best performance, and the variations of GraphFilter with Perplexity, ArmoRM, and AlpaGasus effectively outperform their quality-based counterparts.
> |                  | $\mu_{BENCH}$ | $\mu_{LLM}$ | $\mu_{ALL}$ |
> |------------------|---------------|-------------|-------------|
> | Random           | 47.75         | 41.04       | 45.51       |
> | Perplexity       | 48.27         | 40.28       | 45.61       |
> | ArmoRM           | 48.21         | 42.66       | 46.36       |
> | AlpaGasus        | 48.96         | 41.90       | 46.60       |
> | GraphFilter (Ours)      |               |             |             |
> | with SuperFilter | 50.55         | 42.79       | 47.97       |
> | with Perplexity  | 49.21         | 40.85       | 46.43       |
> | with ArmoRM      | 48.82         | 42.21       | 46.77       |
> | with AlpaGasus   | 49.09         | 42.22       | 46.91       |
>
> **3. Single Visit of N-gram Nodes and Task-Specific Fine-Tuning**
>
> We understand your concern that allowing n-gram nodes to be visited only once might limit the model's ability to learn specific tasks that require repeated exposure to certain n-grams. Our design aims to maximize diversity within a limited budget, but we recognize that this could impact performance on specialized tasks. Although task-specific fine-tuning is beyond the scope of our study, we believe this potential issue can be easily addressed. In our GraphFilter, we consider both quality and diversity during the data selection process. To achieve task-specific data selection, we propose two possible solutions: Firstly, we can utilize the quality metric computed by the task-specific quality estimation model for task-specific re-ranking. Secondly, we introduce a whitelist of task-specific words (e.g., a list of biomedical terminologies), where n-grams containing these words are allowed to be visited multiple times. We will include a discussion in the paper acknowledging this limitation and propose exploring strategies that allow for multiple visits to critical n-grams in future work.
>
> **4. Comparison with SOTA Methods**
>
> In relation to your observation in Section 5.3, we recognize that our method did not surpass state-of-the-art (SOTA) techniques such as DEITA and ARMORM when applied to both instructions and responses. We would like to emphasize that **each method has its own optimal application**, and the sub-optimal performance exhibited by GraphFilter indeed underscores the effectiveness of our design. Furthermore, we present an additional experiment on applying SOTA techniques to instructions only in Point #7, demonstrating that DEITA and AlpaGasus experience performance drops when the responses are not provided to them.

---

> > ### Author Response · Authors · 2024-11-19
> > **Thank You for Your Review + Response (Part 2/2)**
> >
> > **5. Comparison with DEITA in Section 5.2**
> >
> > We appreciate the suggestion to provide a direct comparison with DEITA regarding the complexity, quality, and diversity dimensions. DEITA first sorts the dataset based on the product of the complexity and quality scores and then selects a diverse subset based on the instruction embeddings. If we understand your question correctly, we use the DEITA score (complexity * quality) as the quality metric for GraphFilter and present the result using Llama-3-8B as the model backbone in the following table. We observe that GraphFilter with DEITA complexity * quality performs as well as GraphFilter with SuperFilter and substantially outperforms DEITA. These findings demonstrate that the DEITA score can effectively measure data quality and that GraphFilter is more effective in selecting a diverse subset compared to the embedding-based selection used by DEITA.
> >
> > |                               | $\mu_{BENCH}$ | $\mu_{LLM}$ | $\mu_{ALL}$ |
> > |-------------------------------|---------------|-------------|-------------|
> > | Random                        | 47.75         | 41.04       | 45.51       |
> > | DEITA                         | 48.78         | 41.70       | 46.42       |
> > | GraphFilter (Ours)                   |               |             |             |
> > | with SuperFilter              | 50.55         | 42.79       | 47.97       |
> > | with DEITA complexity * quality | 50.42         | 42.61       | 47.66       |
> >
> > **6. Allowing Multiple Visits to N-gram Nodes**
> >
> > Thank you for your question. We agree that investigating how allowing n-gram nodes to be visited multiple times affects task-specific performance would be insightful. Unfortunately, we are unable to directly conduct additional experiments for this question within the discussion phase because this research question is outside the scope of our study and cannot be quickly implemented. As mentioned in point #3, we propose two possible solutions to address this research question. We will conduct additional experiments to explore this aspect and report on how it influences the model's ability to learn specific tasks or knowledge areas in our future revisions.
> >
> > **7. Applying SOTA Methods Only on Instructions**
> >
> > Your suggestion to evaluate SOTA methods when applied solely to instructions is valuable. We apply several recent SOTA methods to the instructions only and present the results using Llama-3-8B as the model backbone in the following table. We observe that the responses are highly important in determining the data quality, given the observed performance drops. It is important to note that GraphFilter also considers the quality of responses during the data selection process, because SuperFilter measures the data quality based on both the instructions and responses, as shown in Equation 2.
> > |                               | $\mu_{BENCH}$ | $\mu_{LLM}$ | $\mu_{ALL}$ |
> > |-------------------------------|---------------|-------------|-------------|
> > | Random                        | 47.75         | 41.04       | 45.51       |
> > | AlpaGasus                     |               |             |             |
> > | with instruction only         | 48.16         | 41.40       | 45.88       |
> > | with instructions & responses | 48.96         | 41.90       | 46.60       |
> > | DEITA                         |               |             |             |
> > | with instruction only         | 48.45         | 41.66       | 46.22       |
> > | with instruction & responses  | 48.78         | 41.70       | 46.42       |
> > | GraphFilter (Ours)            | 50.55         | 42.79       | 47.97       |

---

> > > ### Comment · Reviewer_PWPP · 2024-11-20
> > >
> > > I thank the authors for their response. Most of my concerns are addressed and I will adjust my score accordingly.

---

### Meta-Review · Area_Chair_smsF · 2024-12-19

**Metareview:**

This paper proposes GraphFilter - a method for selecting high-quality and diverse data for supervised fine-tuning of LLMs. The key idea is to represent the dataset as a bipartite graph linking sentences to n-grams with a priority function that balances between quality and diversity metrics.

Strengths: The approach itself is well-motivated by considering quality and diversity through a graph-based representation. The evaluations are also comprehensive, with multiple models and benchmarks.

Weaknesses: The reviewers raised several issues including the limited exploration of alternative metrics, lack of analysis on the computational complexity and scalability etc.

This paper lies in the borderline from the reviews and also from the internal discussions. While it addresses an important problem and also shows competent performance, the technical contribution falls short of the bar of ICLR. The core approach largely combines existing techniques without substantial innovations, and may not bring too many new insights. Therefore, I'm inclined to reject this paper.

**Additional Comments On Reviewer Discussion:**

All three reviewers rated this paper around the threshold, two of them raised the scores slightly from 5 to 6 during the rebuttal. While comments from Reviewer MtTD seem to be auto-generated, another two reviewers provided detailed feedback and interacted with the authors. However, as mentioned above, the reviewers also expressed their comment that this paper is truly a borderline paper and would benefit from substantial additional development before being ready for publication at ICLR.

---

### Decision · Program_Chairs · 2025-01-22

Reject